# *Curcuma Longa*, the “Golden Spice” to Counteract Neuroinflammaging and Cognitive Decline—What Have We Learned and What Needs to Be Done

**DOI:** 10.3390/nu13051519

**Published:** 2021-04-30

**Authors:** Alessandra Berry, Barbara Collacchi, Roberta Masella, Rosaria Varì, Francesca Cirulli

**Affiliations:** 1Center for Behavioral Sciences and Mental Health, Istituto Superiore di Sanità, Viale Regina Elena 299, 00161 Rome, Italy; barbara.collacchi@iss.it; 2Center for Gender-specific Medicine, Istituto Superiore di Sanità, Viale Regina Elena 299, 00161 Rome, Italy; roberta.masella@iss.it (R.M.); rosaria.vari@iss.it (R.V.)

**Keywords:** turmeric, aging, brain, cognition, bioavailability, oxidative stress, inflammation

## Abstract

Due to the global increase in lifespan, the proportion of people showing cognitive impairment is expected to grow exponentially. As target-specific drugs capable of tackling dementia are lagging behind, the focus of preclinical and clinical research has recently shifted towards natural products. Curcumin, one of the best investigated botanical constituents in the biomedical literature, has been receiving increased interest due to its unique molecular structure, which targets inflammatory and antioxidant pathways. These pathways have been shown to be critical for neurodegenerative disorders such as Alzheimer’s disease and more in general for cognitive decline. Despite the substantial preclinical literature on the potential biomedical effects of curcumin, its relatively low bioavailability, poor water solubility and rapid metabolism/excretion have hampered clinical trials, resulting in mixed and inconclusive findings. In this review, we highlight current knowledge on the potential effects of this natural compound on cognition. Furthermore, we focus on new strategies to overcome current limitations in its use and improve its efficacy, with attention also on gender-driven differences.

## 1. Introduction

Cognitive decline is a highly disabling and prevalent condition in the aging population, greatly affecting physical health and quality of life. Global average life expectancy, as observed in 2019 by the Global Health Observatory (GHO), was estimated to be 73.4 years in the WHO European Region (https://www.who.int/gho/mortality_burden_disease/life_tables/situation_trends_text/en/ accessed on 25 April 2021). In 2050, the number of people over the age of 60 is expected to reach a total of about 2.1 billion (https://www.who.int/ageing/publications/active_ageing/en/ accessed on 25 April 2021). As the ageing population is rapidly growing due to the global increase in life expectancy in westernized life-style countries, the number of people experiencing cognitive impairment is also expected to grow in parallel. Disregarding overt pathologies, the impact of age itself on cognitive abilities is so disruptive and so underestimated that it has been described as “the elephant in the room” [1,2].

The lack of effective pharmacotherapy has led researchers to seek alternative approaches in order to treat or prevent the cognitive decline accompanying ageing. Accumulating evidence suggests that conditions co-occurring in metabolic dysfunctions such as neuroinflammation, oxidative stress (OS), mitochondrial dysfunction or autophagy may all potentially act as triggers for cognitive decline. Indeed, metabolic syndrome (MetS, defined as the presence of three or more of the following five medical conditions: abdominal obesity, high blood pressure, high blood sugar, high serum triglycerides (TG) and low serum high-density lipoprotein—HDL), negatively impacts cognitive performance and brain function possibly increasing neuroinflammation, OS and brain lipid metabolism [3]. Insulin resistance (IR, defined as the inability of peripheral target tissues to respond normally to insulin) is a common condition experienced at old age and often associated with obesity. It typically precedes the onset of type 2 diabetes (T2D) by several years and is considered as a risk factor for cognitive decline in both diabetic and non-diabetic populations [4]. In fact, peripheral IR, by decreasing insulin signaling within the brain, may alter its metabolic functions, increasing OS and neuroinflammation, eventually setting the stage for dementia and neurodegeneration. Thus, neuroinflammation, OS and metabolic dysfunctions involve a strict connection between the brain and the overall metabolic regulation that occur in the periphery. Moreover, changes in microbiota composition and dysbiosis, can potentially influence a number of pathological conditions, include MetS, obesity, T2D, heart failure, and cognitive function (see [5] and references therein). 

While novel target-specific drugs are currently lacking [6], some epidemiological studies indicate that natural antioxidant agents, such as polyphenols, polyunsaturated n-3 fatty acids or vitamin-rich foods may delay the occurrence of neurodegenerative disorders. Polyphenols, in particular (e.g., curcumin and resveratrol) having pleiotropic protective effects appear ideal to prevent or treat conditions (such as AD) whose origin is multifactorial [7]. A growing body of research suggests that regular consumption of natural products (vegetables, fruits, leaves, roots, seeds, berries etc.) rich in polyphenols might improve health outcomes through different mechanisms boosting the organisms’ antioxidant defenses [8]. Natural compounds represent a major source for the discovery of drug targets and are ever increasingly attracting the interest of the scientific community, with the main aim of validating their efficacy for the prevention and treatment of different conditions, including cognitive decline and metabolic disorders [9]. Notwithstanding the growing interest in this class of compounds, rigorous clinic trials addressing their specific effects are lacking or show biases due to the nutritional status of the subjects, genetic background, gender, treatment duration and dose–response relationship [10]. With regard to this latter point, a major drawback is related to their bioavailability, i.e., the amount of compound (or of its active principles) that reaches systemic circulation due to intestinal endothelium absorption and first-pass metabolism. Thus, the use of natural products and nutraceuticals poses important questions regarding human safety and calls for a better understanding of their therapeutic efficacy as well as their mechanisms of action. 

Curcumin, one of the best investigated botanical constituents in the biomedical literature, has been receiving increased interest due to its unique molecular structure, which targets inflammatory and antioxidant pathways, and its potential to improve healthspan [11,12,13,14,15]. The genus Curcuma includes approximately 80 species and is regarded as one of the largest genera of the Zingiberaceae family [16]. Curcumin (1,7-bis(4-hydroxy-3-methoxyphenyl)-1,6-heptadiene-3,5-dione) is a lipophilic polyphenol active extract deriving from the rhizome of *Curcuma longa*. Curcumin (and curcuminoid analogs such as demethoxycurcumin and bisdemethoxycurcumin) provides the characteristic bright yellowish/golden pigment of turmeric widely used in traditional Indian and Chinese medicine from thousands of years because of a number of beneficial effects on human health [17,18]. Today curcumin is used all over the world as a supplement, spice and food additive. It is considered a safe compound suitable for daily dietary use by the United States Food and Drug Administration (FDA), the Joint FAO/WHO Expert Committee on Food Additives (JECFA) and the European Food Safety Authority (EFSA) who have indicated 0–3 mg/kg as an acceptable daily intake (https://www.fda.gov/food/generally-recognized-safe-gras/gras-notice-inventory accessed on 25 April 2021) [19]. Many of its medical uses have been mechanistically validated in in vitro and in vivo preclinical studies (more than 3000 investigations, see [20]) mainly focusing on its antioxidant and anti-inflammatory properties. In recent years, the positive effects of curcumin have been observed in several chronic diseases ranging from cardiovascular, gastrointestinal, neurological disorders and diabetes to several types of cancer [21,22,23,24]. The consumption of curcumin has been associated with a global improvement in the glycemic and lipid profile in patients with MetS [25]. In addition, amelioration in cognitive function in animal models has been widely documented due to its action on structure and functionality of neuronal membranes [26,27]. Despite this evidence it is worth mentioning that curcumin is characterized by poor stability, a feature resulting in an overall low oral absorption, though, once in the blood stream, curcumin appears to be stable and able to reach target tissues [28]. However, as far as the brain is concerned, its application raises the critical issue of its ability to cross the blood–brain barrier (BBB), an issue deserving further investigation [29].

This paper will focus on *Curcuma longa* as a very promising natural compound to counteract inflammaging and cognitive decline. It will review its possible mechanisms of actions and efficacy and will critically address the issue of its bioavailability and describe recent strategies aimed at improving its supplementation.

## 2. Curcumin, Cognitive Decline and Glucose Homeostasis (Peripheral and Central Actions)

The concept of energy homeostasis is receiving much attention nowadays due to the global spread of obesity and diabetes. Hyperglycemia is one of the conditions characterizing MetS, a main risk factor for multi-cause morbidity that includes T2D, cardiovascular disease and also dementia and Alzheimer’s disease (AD). Insulin plays an important role in neuronal survival, in the protection of excitatory synapses and the formation of dendritic spines through the activation of AKT, mTOR and Ras-related pathways that are part of the insulin signaling cascade (see [4] and references therein). Moreover, it regulates levels of GABA, NMDA and AMPA-mediated mechanisms involved in brain plasticity. Insulin binding is highest in the cerebral cortex that plays a role in the control of executive functions as well as in the hippocampus, a brain area involved in learning and memory [4]. The mammalian brain is a highly demanding organ in terms of energy expenditure and shows reduced capacity for cellular regeneration and poor antioxidant defenses, making it particularly susceptible to metabolic and OS insults [30]. Indeed, exposure to hypercaloric diets and obesity has been shown to decrease insulin transport into the mammalian brain, a condition that is restored upon caloric restriction (see [31] and references therein). Recent evidence suggests that IR within the brain, a condition that may be described as the inability of brain cells to respond to insulin and its receptors, has major potential to impact cognitive functions and to contribute to the etiopathogenesis of AD. IR in peripheral tissues and organs (a conditions underlying hyperglycemia and diabetes) is often associated with IR within the brain leading to insulin deficiency and impaired glucose transport inside the neurons [32]. Such a condition may lead to neuronal death, apoptosis and degeneration, predisposing the individual to neurodegenerative diseases and the resultant cognitive decline. Thus, brain desensitization to insulin receptor due to untreated T2D, obesity or chronic consumption of hypercaloric diets may play a key role in what has now been defined as a novel form of diabetes (type 3 diabetes, T3DM) and its complications [33], indicating glucose homeostasis as key in the maintenance of cognitive function.

A growing body of clinical and preclinical data suggests that curcumin holds potential for the control of glucose homeostasis [34] since it may improve glucose uptake, insulin sensitivity and beta islet cell function. Moreover, curcumin may reduce glucose and lipid levels in addition to reducing OS and inflammation [35] by interacting with almost all the players involved in these processes, as demonstrated in in vitro studies [11,35]. A very large amount of literature is now available on curcumin effects. Thus, we will only summarize some of the main aspects.

Preclinical animal models have clearly shown a main effect of natural compounds, including curcumin, on cognitive function during ageing [26,36,37,38,39]. These effects are most likely related to the ability of curcumin to act directly on Aβ plaques as well as to its anti-inflammatory and antioxidant properties. Indeed, a number of preclinical studies have reported downregulation of biomarkers of inflammation (e.g., TNF-α, IL-1β) and OS (e.g., lipid peroxidation, reactive oxygen species—ROS—nitrite and glutathione) believed to be involved in cognitive impairments, confirming the anti-inflammatory and antioxidant properties of curcumin [40,41,42,43,44,45,46,47,48,49,50].

Results of published clinical studies, although in some cases not conclusive, show promise for curcumin’s use as a therapeutic for cognitive decline [51]. The efficacy of curcumin supplementation in humans has been evaluated in several randomized controlled trials, suggesting its potential to reduce blood glucose, C-peptide, glycated hemoglobin (HbA1c), alanine aminotransferase (ALT) and aspartate aminotransferase (AST) in patients with T2D [52]. A recent meta-analysis provides evidence for curcumin’s ability to reduce body mass index, body weight, body fat and leptin values and to increase adiponectin levels in patients with MetS and related disorders [53]. Its effectiveness in reducing TG and C-reactive protein (CRP) and increased adiponectin levels has also been reported [54]. In addition to the ability of curcumin to control glucose homeostasis, and indirectly, to improve/counteract cognitive decline, direct effects have also been observed (and are currently being studied) in the brain. In fact, curcumin has received increased interest due to its unique molecular structure that targets directly amyloid aggregation, one of the major hallmarks of AD. To this regard, Yang and colleagues have provided in vitro and in vivo evidence for curcumin to inhibit Aβ aggregation as well as to prevent oligomer formation [55]; Garcia-Alloza, in a mouse model of AD, found that curcumin reversed existing amyloid pathology and improved the associated neurotoxicity [56]. 

Beyond its use as a compound to prevent/counteract cognitive decline, intriguing evidence supports a possible application for in vivo diagnostics for AD and other related neurodegenerative pathologies [57]. In fact, AD diagnosis can be currently made only by means of clinical criteria supported by invasive and time consuming investigations [58]. Thus, a patient-friendly and repeatable amyloid or Aβ biomarker may alleviate the burden related to currently available diagnostic tools as well as support therapy monitoring in clinical trials [57]. In fact, den Haan and colleagues, by taking advantage of the peculiar feature of curcumin of being naturally fluorescent as well as its Aβ-binding properties, have provided evidence of a selective binding (of curcumin and its related isoforms) to fibrillar Aβ in plaques and cerebral amyloid angiopathy in post mortem AD brain tissues. However, in order to use curcumin as a feasible tool for in vivo detection of Aβ, its poor bioavailability and in vivo metabolism should be carefully considered (see below). 

## 3. Curcumin, Oxidative Stress and Inflammation

The neuropathological features of the brain affected by dementia suggest that the oxidative and inflammatory burden plays a role in the progression of pathological signs by reducing brain plasticity, thus, being an important risk factor for cognitive disability [58]. To this regard, metabolic dysfunctions that are often associated with OS and inflammation, may greatly accelerate the onset and worsen the progression of cognitive functions by promoting brain ageing and reducing healthspan [59]. 

Neuroinflammatory processes are a main feature of neurodegenerative disorders in which microglia and astrocytes are over-activated, resulting in increased production of pro-inflammatory cytokines. Moreover, deficiencies in the anti-inflammatory response may also contribute to neuroinflammation. More specifically, the activated neuroglia by increasing both NF-κB, COX2 and iNOS levels may induce, in turn, the release of pro-inflammatory cytokines, such as IL-6, IL-1α and TNF-β. This pervasive inflammatory condition results in an overall increase in the OS burden leading to neuronal toxicity and the subsequent cognitive deficits characterizing neurodegenerative diseases. Numerous studies have indicated that curcumin is an effective antioxidant both in vivo and in vitro [9]. Curcumin treatment could attenuate cell apoptosis, decrease the level of lipid peroxidation, and increase the activity of various antioxidant enzymes including superoxide dismutase (MnSOD) and glutathione (GSH) [60], thus helping to break the vicious cycle sustaining neuroinflammation and containing the progression of neurodegenerative diseases [61]. The underlying mechanism is possibly associated with the function of NFE2-related factor-2 (Nrf2), a transcription factor promoting the upregulation of antioxidant defenses [61]. 

As far as apoptosis is concerned, many mechanisms have been proposed. Xi-Xun Du and colleagues indicated that curcumin’s property of iron chelation and reduction may underlie its anti-apoptotic effects [62]. Chen and co-workers reported that curcumin may exert its cytoprotective effects against neurotoxic agents via its antiapoptotic and antioxidant properties through the Bcl-2–mitochondrion–ROS–inducible nitric oxide synthase pathway [63]. Moreover, Yu and colleagues reported that the inhibition of JNK pathway and the activation of caspase-3 cleavage might prevent neuronal death [64]. Indeed, the anti-inflammatory and antioxidant properties of curcumin are strictly related to its action on apoptotic pathways and on neuronal death. In fact, pro-inflammatory cytokines are not only involved in the so-called neuroinflammaging but may also trigger the apoptotic process. Likewise, excessive OS may directly lead to mitochondrial swelling and apoptosis. Thus, inflammation and apoptosis are related in a vicious cycle leading to neuronal death [65].

Recently, several studies have highlighted the role of inflammatory pathways mediated by the inflammasome in neurodegenerative diseases. In particular, the NOD-like receptor pyrin domain-containing-3 (NLRP3) has been suggested to play a pathogenic role in several neuroinflammatory diseases, including AD [66]. In vitro and in vivo studies have shown that Aβ peptide activates NLRP3 inflammasome in microglial cells. Furthermore, in a mouse model of AD, NLRP3 knockout (KO) mice were protected from impaired spatial memory performance and showed a decrease in the Aβ plaque load [67], similar results were obtained when a specific NLRP3 inhibitor was administered to mice [68]. This evidence points to the inflammasome as a potential therapeutic target for AD treatment [69]. Notably, recent evidence shows that curcumin, by modulating the activity of NLRP3 inflammasome, could be beneficial in reducing neuroinflammation and/or neurodegeneration in different neurological disorders, such as major depression, brain ischemia, AD and epilepsy [13,70,71].

OS is a condition characterizing aerobic biological systems, the major portion of ROS being generated as a by-product of the electron transport chain operating in the mitochondria [30]. As already mentioned, OS is a condition strongly associated to inflammation that may act both as (con)cause and effect of pathological conditions affecting brain ageing; however, a growing body of evidence suggests that ROS are not only responsible for oxidative damage to cells and macromolecules but they may also play a role as mediators in specific signaling cascades. Hydrogen peroxide (H_2_O_2_) in particular has been identified as a ROS able to affect the ageing process by specifically mediating insulin signaling and promoting fat accumulation, ultimately affecting the ageing process [72]. Worth to notice, the master regulator of this process is the p66Shc *gerontogene*, which, by acting within the mitochondrion, increases the generation of H_2_O_2_, amplifying insulin signaling [73,74,75]. Interestingly, deletion of p66Shc gene in mice resulted in the decreased formation of mitochondrial H_2_O_2_ [75], a feature that has been associated with reduced fat accumulation as well as decreased incidence of metabolic and cardiovascular pathologies [73,76]. Moreover, p66Shc KO mice were characterized by elevated resistance to OS, delayed brain ageing and improved overall healthspan, all features associated with increased brain and behavioral plasticity. In fact, the brain of p66Shc KO mice was characterized by reduced levels of inflammation and OS and increased levels of the neurotrophin brain-derived neurotrophic factor (BDNF); in addition, these mice showed decreased emotionality and improved cognitive function [77,78,79].

Lifestyles have the potential to modulate healthspan during ageing. For example, physical exercise and diet, as well as the consumption of nutraceutical compounds (including curcumin), may greatly contribute to reducing neuroinflammaging by targeting brain pathways related to OS and inflammation (see below, next paragraph). Physical exercise in elderly women has been shown to improve metabolic functions and this was paralleled by a decrease in the peripheral levels of p66Shc gene [80,81]. Very recently, a role for curcumin was also reported in the modulation of the p66Shc gene as its was able to downregulate the expression levels of this gene in peripheral blood mononuclear cells (PBMC), improving diabetic nephropathy in a rat model [82]. These data overall suggest that p66Shc might be exploited as a suitable biomarker of curcumin efficacy to counteract the ageing-related burden and to improve overall healthspan.

## 4. A Potential Mechanism of Action: Curcumin as a “Hormetin”

A large epidemiological Indo-US Cross National Dementia study showed that rural Indian populations have a low prevalence of AD and AD-associated dementia compared to the US population, and this may be linked to the high curcumin consumption [17], although such correlation does not necessarily imply causative connection.

An ever-increasing body of evidence suggests that natural compounds may alleviate the burden of chronic diseases by increasing individuals’ ability to cope with OS. Notably, differently from the consumption/administration of antioxidants (of natural or synthetic origin), whose beneficial effects are still debated, most natural compounds do not act solely as free radical scavengers but rather, and most interestingly, as “antioxidant boosters” [83]. In this regard, it is important to point out that ROS also function as signaling molecules underlying physiological processes and, for this reason, their generation and scavenging needs to be tightly regulated (see below). 

As reviewed by Lee and colleagues [8], cellular stressors that are relevant to the pathogenesis of chronic diseases may be roughly categorized into four general types: (1) OS resulting from the unbalance between ROS production and the organism’s antioxidant defenses; (2) metabolic stress deriving from impaired cellular bioenergetics and mitochondrial dysfunctions; (3) proteotoxic stress protein misfolding and aggregates accumulation, and (4) inflammatory stress that leads to the production of ROS from immune cells. All these stressors are relevant for brain ageing. Moreover, and most importantly, OS is also involved as a cause or as a consequence in all the above-mentioned categories. Cytotoxic effects of ROS contribute to the death of neurons during chronic neurodegenerative diseases such as AD and Parkinson’s disease and also during the ageing process. However, ROS also function as signaling molecules underlying physiological processes including cell proliferation, migration, and survival through the regulation of neurotrophic factors [84,85]. Therefore, the generation and scavenging of ROS needs to be tightly regulated and nutraceutical compounds appear to be good candidates to play a role in this process. In this regard, the cells throughout the body and brain might trigger stress signaling pathways, eventually leading to the enhancement of their own resistance to further stressors, including OS. An intriguing hypothesis suggests that nutraceuticals might be perceived as potentially toxic by the organism (at high doses). However, exposure to low doses of these compounds might stimulate the organism’s hormetic/adaptive responses aimed at counteracting such threats (“hormesis hypothesis”, see [8]. Indeed, a recent paper by Calabrese and co-workers provides evidence that curcumin displays hormetic-like biphasic dose response features that are independent from the biological model used for investigation, cell type, and endpoints [86]. These findings hold major implications for study design, including selection of doses and sample size, also considering the specific context of bioavailability and pharmacokinetics, see Figure 1.

## 5. Criticisms to Be Considered in Curcumin Supplementation

Despite plenty of data on the positive health effects of curcumin [88,89], some problems strongly limit its effectiveness and usefulness. First of all, its low bioavailability. Curcumin is characterized by low water solubility and high instability in most body fluids; in addition, it is poorly absorbed by the gastrointestinal tract. In fact, curcumin is rapidly metabolized by the large intestine and by liver enzymes, leading ultimately to the production of sulphate and glucuronide O-conjugated metabolites [90]. Specifically, curcumin that does reach the blood flow undergoes phase I (reduction) and phase II metabolism (conjugation). Reductases reduce curcumin to dihydrocurcumin, tetrahydrocurcumin and hexahydrocurcumin (phase I) [91,92], then these phase I metabolites are conjugated to sulphates and glucuronides (phase II) [93,94].

In animal and human studies, a low concentration of curcumin in blood plasma and urine was observed after oral administration, in particular, serum curcumin levels are undetectable in humans even after high oral doses (up to 8 g/day) [95,96]. Finally, the presence of curcumin in the blood is not sufficient to ensure the delivery in the brain to exert the neuroprotective activity because several studies have demonstrated that it does not easily cross the BBB. All these factors together have pushed research towards finding new formulations or new ways of administration able to stabilize the molecule and increase its bioavailability, by reducing its metabolism and increasing the retention time in the bloodstream [97]. To this regard, the simultaneous administration of curcumin with piperine [98,99], essential oil [100] or milk [101] have been suggested to stimulate the gastrointestinal system, prevent the efflux of curcumin and to increase absorption and metabolism. Many studies have been aimed at devising and testing new drug delivery strategies using, for example, carriers such as soy lecithin phosphatidylcholine (phytosome, Meriva^®^) that improve both the absorption of curcumin in the intestine as well as its penetration into the cells [102,103,104,105]. In addition, nanoparticles, liposomes, micelles, phospholipid complexes, emulsions, microemulsions, nanoemulsions, solid lipid nanoparticles, nanostructured lipid carriers, biopolymer nanoparticles and microgels are able to increase curcumin bioavailability by enhancing small intestine permeation, preventing possible degradation in the microenvironment, eventually increasing plasma half-life and enhancing curcumin efficacy [106,107,108,109,110,111,112,113]. Another way to administer curcumin is by using exosomes, nanovesicles (30–100 nm) that are generated within the cell in the endosomal network. This method, which appears to be safe and non-cytotoxic [114,115], increases both the plasma concentration as well as the bioavailability of curcumin 5–10 times more than curcumin alone [116]. Other studies have confirmed that increased solubility, stability and bioavailability of curcumin can be obtained by incorporating it into exosomes. All these different curcumin formulations enhance its bioavailability and allow for greater persistence in the body, better permeability and resistance to metabolic processes and a higher efficacy [97]. Other studies have focused on changing the chemical structure of curcumin, generating curcumin derivatives that may show not only an improved pharmacological activity, but also better physicochemical and pharmacokinetic properties [117,118], see Figure 2.

It is worth noting that the paradox of the curcumin pharmacological effect, despite its poor bioavailability could be, at least partially, explained by the influence of the microbiome on curcumin metabolism. Indeed, the biological activities of curcumin are linked to the digestion by the intestinal flora, which can produce active metabolites. Thus, the beneficial effects of curcumin seem to depend on the individual ability to metabolize it, that is, to the composition of each person’s intestinal microbiota [120]. Several microorganisms capable of modifying curcumin have been identified in the human microbiota, including *Bifidobacteria longum*, *Bifidobacteria pseudocatenulaum*, *Enterococcus faecalis*, *Lactobacillus acidophilus* and *Lactobacillus casei* [121]. Many of them metabolize curcumin to a large extent (more than 50%) and produce a number of metabolites (approximately 23 of them have been identified) by different metabolic pathways such as acetylation, hydroxylation, reduction, demethylation [122]. Several studies have shown that the metabolites have similar properties to curcumin [123] and many of them have been shown to exhibit neuroprotective effects [124,125], suggesting that curcumin transformed by the gut microbiota could be useful for microbiome-targeting therapies for AD. In fact, bidirectional communication exists between the central nervous system and the gut microbiota, which plays a key role in human health [5]. A growing body of evidence suggests that the gut microbiota can influence human brain and behavior, and the different metabolites secreted by the gut microbiota can affect the cognitive abilities of patients diagnosed with neurodegenerative disorders [126,127]. Changes in the gut microbiota composition, caused by dietary habits, antibiotic exposure and/or infections might result in conditions of dysbiosis (also known as dysbacteriosis) that are involved in the etiopathogenesis of different diseases in humans, including MetS, obesity, T2D and neurodegenerative disorders [127,128]. Indeed, changes in gut microbiota homeostasis leads to increased intestinal permeability that, in turn, results in the translocation of bacteria and endotoxins across the epithelial barrier, a condition that might trigger an immunological response associated with the production of pro-inflammatory cytokines. Such mechanisms have the potential to greatly contribute to neuroinflammation through the secretion of pro-inflammatory cytokines, ultimately altering brain functions [5]. Thus, curcumin supplementation may be useful to counteract/prevent cognitive decline also by mechanisms involving the preservation of individuals’ microbiota homeostasis. 

## 6. What have We Learned and What Needs to Be Done

Although only a few clinical studies have examined curcumin’s effect on human cognitive functioning, the results of these trials are sometimes inconsistent, highlighting the difficulty in translating basic research to the clinic. While some studies report no cognitive enhancing effects of curcumin [129,130], other data indicate positive effects of this compound on cognitive function [131,132,133]. Also in preclinical studies, we found great heterogeneity and limited ability to standardize age of administration or endpoints [39,73,134,135].

Nonetheless, even in very different animal models, curcumin consistently decreases both systemic and central neuroinflammation while improving redox state [51]. As many clinical studies did not include inflammatory and oxidative biomarkers, future trials should target these measures to yield further insight into why curcumin has not shown consistent cognitive effects in humans. Among these, changes in p66Shc transcription in PBMC might be exploited as a potential biomarker of curcumin efficacy [82,136]. Bioavailability may be one main factor that increases variability between studies. Administration of any of the three constituents (curcumin, bisdemethoxycurcumin and demethoxycurcumin) separately instead of the parent curcuminoid mixture was recommended as a more efficient way of treatment [125]. Furthermore, a synergistic effect of curcumin with other dietary supplements, such as piperine, α-lipoic acid, N-acetylcysteine, B vitamins, vitamin C, and folate can improve its effects [137,138]. Nowadays, nanoparticles are mainly used since they demonstrate better penetration at the BBB and cause more evident biochemical changes than free curcumin [9,46,49,139,140].

Nevertheless, there is still room for improvement. As an example, use of soy lecithin phosphatidylcholine (phytosome, Meriva^®^), which improves both the absorption of curcumin in the intestine as well as its penetration into the cells, has been used and has resulted in significant effects in clinical studies assessing a number of variables such as liver function, inflammation, gastrointestinal disturbances and T2D [141,142,143,144,145]. Thus, using new delivery systems that increase bioavailability appears to be a *must-do* for future clinical trials [26,51].

Other drawbacks of previous clinical trials are limited power, different formulations of curcumin used [129] and differences in ethnicity, as an example Caucasians vs. Asians, making it difficult to compare results derived from different studies [146]. Genetic factors, such as polymorphisms or dietary differences, such as higher consumption of curcumin in Asia, could underlie differential effects in different ethnic groups. 

A dose–response relationship should also be taken into account. The optimal dose would have maximum cognitive enhancing effects with the safest pharmacokinetic profile. We have previously shown in animal studies that, in general, beneficial effects of natural compounds on cognition are dose-dependent with the higher dosages generally being more effective compared to lower dosages, although a significant effect of nutraceutical compounds on memory retention and OS was also demonstrated at the lowest dose [39]. In addition, when selecting the most effective dose in basic or clinical studies it should be carefully taken into account that curcumin commonly displays a biphasic dose–response curve, such as hormetic compounds do [86,147]. In this regard, it should be noted that problems such as optimal dose and bioavailability are common to different natural compounds. A recent review by Mazzanti and Di Giacomo pointed out that such issues may also represent a major drawback when trying to compare the efficacy of curcuma and resveratrol, two polyphenols with very similar antioxidant and anti-inflammatory properties, to counteract cognitive decline [7]. As an example, both curcumin and resveratrol are able to activate the Nrf2 and NF-kB pathways as well as to modify insulin signaling. Their effects mostly overlap and both have been shown to enhance cognitive function in animal models [8]. More recently, Saleh and co-workers when trying to compare the protective effects of curcumin, resveratrol and sulphoraphane in an in vitro study, confirmed that enhancing the delivery of phytochemicals (either by designing novel nanostructures or using mixtures of natural compounds that work synergistically) is a priority in this field of research [148]. Thus, further research is mandatory to establish the most effective substance to be used, in which conditions and for whom.

## 7. Targeting Both Genders

The male-female health-survival paradox, also known as the morbidity-mortality paradox or gender paradox, poses that women live longer, though they experience more disabilities and medical issues throughout life, compared to men. Such a paradox holds true in almost every country in the world since virtually all the primary causes of death are higher for men at all ages [149]. Thus, due to their greater resilience to stress, women live longer than men but experience higher rates of physical illness, leading to debilitating, though rarely lethal, conditions. Several hypotheses have been proposed for this phenomenon that could be interpreted as a sex-driven resilience to stressors (leading to longer lifespan), including more efficient female immune functioning, the protective role of estrogens as well as increased antioxidant capacity [149]. 

Many chronic conditions, including dementia and AD, though not lethal *per se*, are strongly linked to disability and loss of physiological functions. Alzheimer’s disease is a multifactorial neurodegenerative disorder, the development of which depends upon both environmental as well as genetic risk factors. A recent review by Christensen and Pike put the attention specifically on two risk factors that may play a key role in the initiation and/or progression of AD and that may be particularly problematic for women: inflammation and obesity [150]. Alzheimer’s disease and other dementias are highly prevalent among women [151] and the onset of menopause, which is associated with estrogen breakdown, decreases women’s vulnerability threshold to both metabolic and cognitive disorders that depend upon central adiposity and overall inflammation [150].

In women, the shift from adulthood to middle-age is characterized by an overall increase in the proportions of overweight and/or obese subjects [152]. Clinical and preclinical studies suggest that the age-related loss of ovarian hormones results in weight gain and contributes to changes in the distribution of adipose tissue leading to increased waist-to-hip ratio [153]. Elevated adiposity increases the risk of different pathological or sub-pathological conditions, including MetS [154], T2D [155] and AD [156]. One consequence of increased adiposity that may underlie its pathogenic role is chronic inflammation, which is observed both systemically as well as in the brain (see [150] and references therein). A large body of data supports the link between estrogens (and other sex steroid hormones) and the modulation of the individual inflammatory profile. Indeed, estrogens are powerful anti-inflammatory mediators, thus, the drop experienced by women at menopause triggers a rise in pro-inflammatory cytokines that may place tissues throughout the body at increased risk of inflammaging-associated diseases [157]. A recent clinical trial aimed at assessing the efficacy of oral curcumin (500 mg) administration twice/day for 8 weeks on anxiety and other specific symptoms that accompany menopause, has shown that curcumin significantly reduced hot flashes in postmenopausal women [158]. In this context, it is possible to foresee the use of curcumin as a possible preventive strategy to be administered in pre-menopausal women aimed at boosting anti-inflammatory and antioxidant capacity when natural defenses that women are endowed with start to be threatened by the absence of estrogens (see Figure 3). In fact, it is important to stress here that, given the current knowledge on age-associated cognitive decline, it would be unrealistic to foresee a therapeutic use of nutraceuticals in overt pathologies characterized by massive neurodegeneration, such as AD. Thus, early diagnosis of the pathology and prompt interventions should be aimed at preventing or slowing down its progression. In this regard, the identification of suitable, least-invasive biomarkers (e.g., through blood tests) should be considered a priority. Moreover, it is more and more evident that prevention through specific diets and physical exercise will represent the key in the future to counteract cluster conditions that put the individual at greater risk for cognitive and physical decline, including, e.g., MetS [39].

## 8. Conclusions

In this review, we have touched upon a number of critical issues that should be taken into account when designing preclinical and clinical studies aimed at assessing curcumin efficacy on cognitive functions. Among these, one important factor that clearly needs to be tackled in the future has to do with increasing its bioavailability as well as controlling the impact of nutritional status/diet and lifestyle on curcumin’s effects. Diet is also likely to influence microbiota status, thus, controlling for nutritional status will be crucial for effective future studies.

Another important point relates to the need to assess the effects of this natural compound on both males and females, both in preclinical and clinical studies. As dementia and AD show a much greater prevalence in the female population, it is imperative to address this issue by targeting both sexes/genders [151].

Moreover, given the overlap between the mechanisms of action of many compounds (see, e.g., curcumin and resveratrol), it will be important to target multiple molecular pathways to maximize the effects. Traditional medicines mostly use mixtures of phytochemicals, rather than individual compounds, which suggests the need to examine, with rigorous clinical trials, the role of plant mixtures on brain health [9,160].

Ultimately, in order to assess the effects of natural compounds, such as curcuma, we need to refine our ability to measure health (and the lack of) in a life-long perspective and to characterize the conditions for the transition from health to disease [39,73,135].

## Figures and Tables

**Figure 1 nutrients-13-01519-f001:**
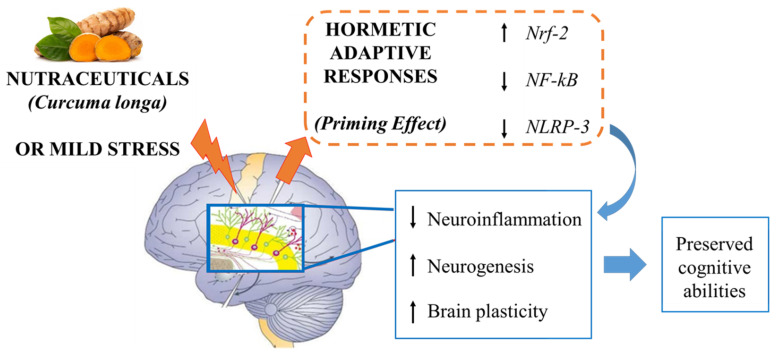
An intriguing hypothesis suggests that nutraceuticals might be perceived as potentially toxicant by the organism (at high doses), however, exposure to low doses of these compounds might stimulate the organism’s hormetic/adaptive responses aimed at counteracting such threats. Likewise, *Curcuma longa* acting as a mild stress might trigger signaling cascades boosting antioxidant and anti-inflammatory pathways (related to Nrf2, NF-kB, NLRP3), leading to decreased neuroinflammation, increased neurogenesis and brain plasticity, finally improving cognitive abilities. Figure adapted with permission from [87], Copyright © 2014 University of Massachusetts.

**Figure 2 nutrients-13-01519-f002:**
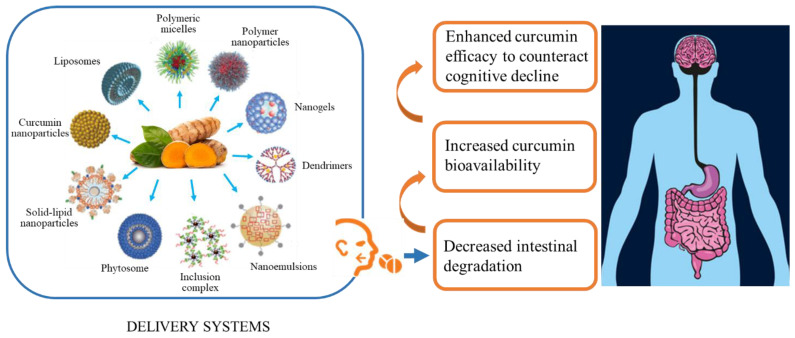
New delivery systems such as polymeric micelles, polymer nanoparticles, nanogels, dendrimers, nanoemul-sions, inclusion complexes, phytosomes, solid-lipid nanoparticles, curcumin nanoparticles and liposomesnanoparticles, liposomes, micelles, nanogel, dendrimers, nanoemulsions, inclusion complexes and phytosomes have the potential to reduce intestinal degradation and increase curcumin bioavailability, ultimately enhancing its efficacy throughout the body and the brain. Within this latter organ the increased curcumin bioavailability might counteract cognitive decline. Figure adapted with permission from [119]. Copyright © 2018, Said Moselhi.

**Figure 3 nutrients-13-01519-f003:**
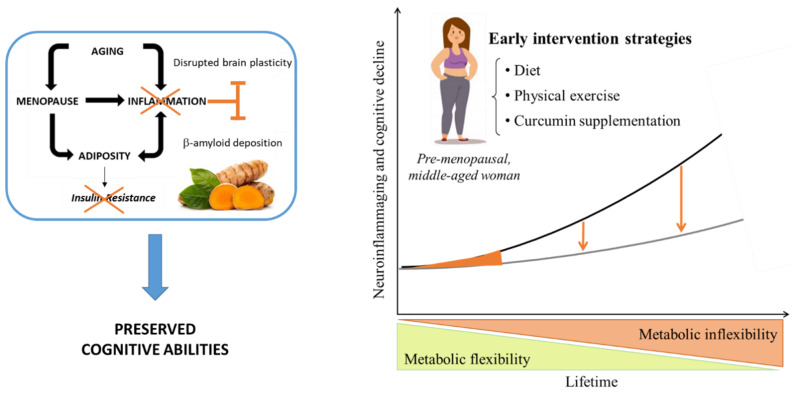
During middle-age, women experience menopause that is characterized by the physiological drop in the protective effects of estrogen hormones. Such a drop leads to increased inflammation and adiposity, two conditions that might reinforce one another in either decreasing brain plasticity or increasing amyloid deposition or both. Curcumin administration in middle-aged pre-menopausal women has the potential to break this vicious cycle overall boosting anti-inflammatory defenses, counteracting fat-mediated insulin resistance and preserving brain plasticity (left panel). So far, no specific sex-differences have been found with regard to curcumin dose–response efficacy. However, it is important to stress that identifying sex-specific critical time windows throughout life to start curcumin administration might be equally important and might improve the chance to protect the brain and to counteract cognitive decline. Such a time window in women might be the middle-age, right before the beginning of menopause, when the organism still retains a certain degree of metabolic flexibility. Thus, the earlier the intervention (also through specific diets and physical exercise) the greater the chance to prevent the decay in mental and physical health (**right panel**). Figure adapted with permission from [150], Copyright © 2015 Christensen and Pike, and from [136,159]: © 2007 Springer International Publishing and © 2020 Published by Elsevier Ltd on behalf of IBRO.

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
