# Peer review of "Curcuma Longa, the “Golden Spice” to Counteract Neuroinflammaging and Cognitive Decline—What Have We Learned and What Needs to Be Done"

_nutrients, 2021, doi:10.3390/nu13051519_

Round 1
Reviewer 1 Report
This is an extensive review on the potential health benefits of curcuma longa written by experts in the field. I miss two things: a comparison of curcuma's impact on neuroinflammtion and cognitive function with that of other antioxidants such as resveratrol (i.e. what substance is most effective) and some debate of the value of antioxidants in general (s. Scudellari, Nature, 2015). Minor: the cerebral cortex is not a brain area (p. 3), it should read "meal rich in fat" (p. 11).
Author Response
REVIEWER #1
This is an extensive review on the potential health benefits of curcuma longa written by experts in the field.
I miss two things: a comparison of curcuma's impact on neuroinflammation and cognitive function with that of other antioxidants such as resveratrol (i.e. what substance is most effective).
We thank the reviewer for the suggestion. The paper has been modified as follows: Pag. 2 The mechanism of action of curcuma is most likely shared by other compounds. Polyphenols, in particular (eg.: curcumin and resveratrol) having pleiotropic protective effects which appear ideal to prevent or treat disorders (such as AD) whose origin is multifactorial (Mazzanti and Di Giacomo, 2016)”.
Pag. 10 “To this regard, it should be noticed that problems such as optimal dose and bioavaila-bility are common to different natural compounds. A recent a review by Mazzanti and Di Giacomo has pointed out that such issues may also represent a main drawback when trying to compare the efficacy of curcuma and resveratrol - two polyphenols with very similar antioxidant and anti-inflammatory properties - to counteract cognitive decline (Mazzanti and Di Giacomo, 2016). As an example, both curcumin and resveratrol are capable to activate the Nrf2 and NF-kB pathways as well as to modify insulin signalling. Their effects mostly overlap and both have been shown to enhance cognitive function in animal models (Lee and Mattson 2014). More recently Saleh and co-workers when trying to com-pare the protective effects of curcumin, resveratrol and sulphoraphane - in and in vitro study – confirm that enhancing the delivery of phytochemicals (either by designing novel nanostructures or using mixtures of natural compounds that work synergistically) is a priority in this field of research (Saleh et al., 2021). Thus, further research is mandatory to establish the most effective substance to be used in which conditions and for whom”.
….and some debate of the value of antioxidants in general (s. Scudellari, Nature, 2015).
We thank the reviewers for pointing out this issue. The text on pag. 6 has been implemented as follows “Worth to notice, differently from the consumption/administration of antioxidants (of natural or synthetic origin), whose beneficial effects are still debated, most natural compounds do not act solely as free radical scavengers but rather, and most interesting-ly, as “antioxidant boosters” (Scudellari, 2015). To this regard, it is important to point out that ROS also function as signalling molecules underlying physiological processes and, for this reason, their generation and scavenging needs to be tightly regulated (see be-low)”.
Minor: the cerebral cortex is not a brain area (p. 3)
We thank the reviewer for pointing out this issue. The sentence has been changed as follows: “Insulin binding is highest in the cerebral cortex that plays a role in the control of executive functions as well as in the hippocampus a brain area involved in learning and memory (Nguyen et al. 2020)”.
….it should read "meal rich in fat" (p. 11).
The typo has been corrected.
Reviewer 2 Report
The authors performed a review article to illustrate the effects of curcumin on neuroinflammaging and cognitive decline. Basically, the manuscript is well written, some suggestions as below:
- The authors write: “This paper will focus on Curcuma Longa as a very promising natural compound to counteract inflammaging and cognitive decline. It will review its possible mechanisms of actions and efficacy by taking into account the role of intestinal microbiota and will critically address the issue of its bioavailability and describe recent strategies aimed at improving its supplementation.” But in the article, we can’t read the detail of the mechanisms that how curcumin interacts with the ingested-microorganism probiotics (i.e., ingested microorganisms associated with beneficial effects for the host). What kinds of species? What are the biological effects? These points should be addressed structurally.
- The authors write: “Curcumin treatment could attenuate cell apoptosis, decrease the level of lipid peroxidation, and increase the activity of various antioxidant enzymes including superoxide dismutase (MnSOD) and glutathione (GSH) [59] helping to break the vicious cycle sustaining neuroinflammation and containing the progression of the neurodegenerative diseases” The article should address some mechanism about 1)how curcumin attenuates cell apoptosis? 2) the relationship between apoptosis, inflammation, and neuroinflammaging.
- What is the effects dose of curcumin to prevent diseases? What are your opinions?
- In the conclusions parts, the authors should illustrate the final summary of all the information instead of others studies.
Author Response
REVIEWER #2
The authors performed a review article to illustrate the effects of curcumin on neuroinflammaging and cognitive decline. Basically, the manuscript is well written, some suggestions as below:
The authors write: “This paper will focus on Curcuma Longa as a very promising natural compound to counteract inflammaging and cognitive decline. It will review its possible mechanisms of actions and efficacy by taking into account the role of intestinal microbiota and will critically address the issue of its bioavailability and describe recent strategies aimed at improving its supplementation.” But in the article, we can’t read the detail of the mechanisms that how curcumin interacts with the ingested-microorganism probiotics (i.e., ingested microorganisms associated with beneficial effects for the host). What kinds of species? What are the biological effects? These points should be addressed structurally.
We thank the reviewer for pointing out this issue which is very important. Given the focus of the review, however, a detailed discussion on the interaction between curcuma and the intestinal microbiota appears beyond its scope. The role of gut microbiota in this context should be intended only to partially explain the paradox of the curcumin pharmacological effect, despite its poor bioavailability (see sentence pag. 8 “It is worth to notice that the paradox of the curcumin pharmacological effect, despite its poor bioavailability, could be, at least partially, explained by the influence of the microbiome on curcumin metabolism”). For this reason, the final sentence of the introductory paragraph has been toned down removing emphasis on microbiota. See the following: “It will review its possible mechanisms of actions and efficacy and will critically address the issue of its bioavailability and describe recent strategies aimed at improving it supplementation”. Moreover, in this same issue of Nutrients the review by Scazzocchio, B., Minghetti, L., D’archivio, M., 2020 (Interaction between gut microbiota and curcumin: A new key of understanding for the health effects of curcumin. Nutrients. https://doi.org/10.3390/nu12092499) is present and addresses all the points raised by the reviewer extensively. The review by Scazzocchio et al. has been cited in the text and pointed out as a reference for further details on this specific topic.
1) The authors write: “Curcumin treatment could attenuate cell apoptosis, decrease the level of lipid peroxidation, and increase the activity of various antioxidant enzymes including superoxide dismutase (MnSOD) and glutathione (GSH) [59] helping to break the vicious cycle sustaining neuroinflammation and containing the progression of the neurodegenerative diseases” The article should address some mechanism about how curcumin attenuates cell apoptosis?
We thank the reviewer for the thorough question. The following paragraph has been added to the text (pag 5) “As far as apoptosis is concerned, many mechanisms have been proposed. Xi-Xun Du and colleagues propose that curcumin property of iron chelation and reduction may underlie its anti-apoptotic effects (Xi-Xun Du 2012). Chen and co-workers report that curcumin may exert its cytoprotective effects against neurotoxic agents via its anti-apoptotic and antioxidant properties through the Bcl-2–mitochondrion–ROS–inducible nitric oxide synthase pathway (Chen et al. 2006). Moreover, Yu and colleagues reported that the inhibition of JNK pathway and the activation of caspase-3 cleavage might prevent neuronal death (Yu et al. 2010)”.
2) The relationship between apoptosis, inflammation, and neuroinflammaging.
We thank the reviewer for the thorough question. The following paragraph has been added to the text (Pag 5) “Indeed, the anti-inflammatory and anti-oxidant properties of curcumin are strictly related to its action on apoptotic pathways and on neuronal death. In fact, pro-inflammatory cytokines are not only involved in the so-called neuroinflammaging but may also trigger the apoptotic process. Likewise, excessive OS may directly lead to mitochondrial swelling and apoptosis. Thus inflammation and apoptosis are related in a vicious cycle leading to neuronal death (Guo et al. 2020)”.
3) What is the effects dose of curcumin to prevent diseases? What are your opinions?
We thank the reviewer for rising up this point that gives us the opportunity to clarify that, so far, this is still an important matter of investigation, mainly related to the bioavailability of curcuma (and of many other natural compounds). The following sentence has been added to the text: Pag. 10 “To this regard, it should be noticed that problems such as optimal dose and bioavaila-bility are common to different natural compounds. A recent a review by Mazzanti and Di Giacomo has pointed out that such issues may also represent a main drawback when trying to compare the efficacy of curcuma and resveratrol - two polyphenols with very similar antioxidant and anti-inflammatory properties - to counteract cognitive decline (Mazzanti and Di Giacomo, 2016). As an example, both curcumin and resveratrol are capable to activate the Nrf2 and NF-kB pathways as well as to modify insulin signalling. Their effects mostly overlap and both have been shown to enhance cognitive function in animal models (Lee and Mattson 2014). More recently Saleh and co-workers when trying to com-pare the protective effects of curcumin, resveratrol and sulphoraphane - in and in vitro study – confirm that enhancing the delivery of phytochemicals (either by designing novel nanostructures or using mixtures of natural compounds that work synergistically) is a priority in this field of research (Saleh et al., 2021). Thus, further research is manda-tory to establish the most effective substance to be used in which conditions and for whom”. Moreover, as already mentioned in the text “Authority (EFSA) have indicated 0-3 mg/kg as an acceptable daily intake (https://www.fda.gov/food/generally-recognized-safe-gras/gras-notice-inventory) (Sharifi-Rad et al., 2020)”.
4) In the conclusions parts, the authors should illustrate the final summary of all the information instead of others studies.
We thank the review for this comment. The conclusions have been restructured to include final remarks and main points for future work, see the following: “In this review, we have touched upon a number of issues that should be taken into account when designing preclinical and clinical studies aimed at assessing curcumin efficacy on cognitive functions. Among many, one important factor that clearly needs to be tackled in the future has to do with increasing its bioavailability as well as controlling the impact of nutritional status/diet and lifestyle on curcumin’s effects. Diet is also likely to influence microbiota status thus controlling for nutritional status will be crucial for effective future studies.
Another important point relates to the need of assessing the effects of this natural compound on both males and females both in preclinical and clinical studies. As dementia and AD show a much greater prevalence in the female population, it is imperative to address this issue by targeting both sexes/genders (2020 Alzheimer's disease facts and figures. Alzheimer's Dement., 16: 391-460. https://doi.org/10.1002/alz.12068).
Moreover, given the overlap between the mechanisms of action of many com-pounds (see e.g. curcumin and resveratrol), it will be important to target multiple molecular pathways to maximize the effects. Traditional medicines use mostly mixtures of phytochemicals, rather than individual compounds, which hints to the need to examine, with rigorous clinical trials, the role of plant mixtures on brain health (Long et al., 2015; Tewari et al., 2018).
Ultimately, in order to assess the effects of natural compounds, such as curcuma, we need to refine our ability to measure health (and the lack of) in a life-long perspective and to characterize the conditions for the transition from health to disease (Berry and Cirulli, 2013; Cohen et al., 2019; Musillo et al., 2021)”